# Polyether Ether Ketone Coated with Ultra-Thin Films of Titanium Oxide and Zirconium Oxide Fabricated by DC Magnetron Sputtering for Biomedical Application

**DOI:** 10.3390/ma15228029

**Published:** 2022-11-14

**Authors:** Igor O. Akimchenko, Sven Rutkowski, Tuan-Hoang Tran, Gleb E. Dubinenko, Vsevolod I. Petrov, Anna I. Kozelskaya, Sergei I. Tverdokhlebov

**Affiliations:** 1Weinberg Research Center, School of Nuclear Science & Engineering, Tomsk Polytechnic University, 30, Lenin Avenue, Tomsk 634050, Russia; 2Tomsk Scientific Center of the Siberian Branch of the Russian Academy of Sciences, 10/4, Akademicheskii Prospekt, Tomsk 634055, Russia

**Keywords:** ultra-thin films, PEEK, biocompatible coatings, DC magnetron sputtering, titanium, zirconium

## Abstract

Recently, polyether ether ketone has raised increasing interest in research and industry as an alternative material for bone implants. This polymer also has some shortcomings, as it is bioinert and its surface is relatively hydrophobic, causing poor cell adhesion and therefore slow integration with bone tissue. In order to improve biocompatibility, the surface of polyether ether ketone-based implants should be modified. Therefore, polished disc-shaped polyether ether ketone samples were surface-modified by direct current magnetron sputtering with ultrathin titanium and zirconium coatings (thickness < 100 nm). The investigation results show a uniform distribution of both types of coatings on the sample surfaces, where the coatings mostly consist of titanium dioxide and zirconium dioxide. Differential scanning calorimetry revealed that the crystalline structure of the polyether ether ketone substrates was not changed by the coating deposition. Both coatings are amorphous, as shown by X-ray diffraction investigations. The roughness of both coating types increases with increasing coating thickness, which is beneficial for cell colonization. The coatings presented and investigated in this study improve wettability, increasing surface energies, in particular the polar component of the surface energies, which, in turn, are important for cell adhesion.

## 1. Introduction

Polyether ether ketone (PEEK) is a high-performance thermoplastic polymer that is increasingly being used in orthopedics [1]. Compared to the elastic modulus of titanium (Ti; 102–110 GPa) [2], PEEK has a relatively low modulus of elasticity (3–5 GPa), which is closer to the diaphysis of a bone [3]. Such a low modulus value should prevent the occurrence of high strain/stress areas at the bone-implant interface, which can complicate bone reconstruction. In addition, PEEK does not cause artifacts in computed tomography (CT) scans and magnetic resonance imaging (MRI) [4]. PEEK shows resistance to in vivo degradation; thus, several orthopedic and spinal devices made from PEEK have been approved by the US Food and Drug Administration (FDA) [5] (p. 444). Moreover, PEEK has been commercially available as a biomaterial for long-term implants since 1998 [6]. It has also been shown that PEEK is a much easier material to process than metals and metal alloys in terms of manufacturing, processability, costs and ability to be easily 3D printed [7,8]. Thus, PEEK is a highly effective thermoplastic polymer to replace (or as an alternative to) metal implants in the field of orthopedics. On the other side, due to its hydrophobic properties, PEEK has a low surface energy that limits cell adhesion; hence, PEEK is a biologically inert material in terms of biocompatibility and osseointegration [9]. Surface properties of implants are important for tissue response. Therefore, modification of the PEEK surface can make it more attractive for osteoblast growth, leading to better integration with bone tissue.

Various methods are used to modify the surface. Since PEEK has a high chemical resistance, chemical modification of the surface of PEEK is a difficult task that can be performed with highly concentrated inorganic acids [10,11]. Therefore, the most common modification strategies involve physical/chemical surface modification and/or coating of PEEK. Nowadays, much attention is paid to the application of bioactive coatings. The surface of PEEK can be coated with various materials, including hydroxyapatite (HA), titanium (Ti), titanium dioxide (TiO_2_), titanium nitride (TiN), gold (Au), etc. [12,13,14,15,16].

HA is the most popular PEEK coating material. In general, HA is a widely used calcium phosphate-based bioceramic, which is the most similar synthetic analog of the human bone mineral [17] (pp. 139–171). HA significantly improves the bioactivity of PEEK in terms of cell adhesion, morphology and proliferation [18], and thus the osseointegration is increased for HA-coated PEEK compared to uncoated PEEK [19]. However, PEEK is inherently inert and has a higher thermal expansion coefficient (5.8 × 10^−5^ °C) than HA (1.4 × 10^−5^ °C) [20,21], resulting in low adhesion of the HA film to PEEK.

In addition, coatings consisting of Ti and its compounds TiO_2_ and TiN are used to improve the biocompatibility and bioactivity of the PEEK surface. Chang Yao et al. showed that Ti-coating on PEEK significantly increased osteoblast adhesion and improved their proliferation compared to uncoated PEEK [22]. In Ref. [23], a nanoporous TiO_2_ coating immobilized with the bone morphogenetic protein BMP-2 and applied to PEEK showed in in vitro and in vivo studies a significant improvement in adhesion, proliferation and differentiation of the osteoblast precursor cell line MC3T3-E1, which means that the osteoconductivity was increased. Various review articles also show that Zr-based implants have a comparable or even better healing response and a less inflammatory effect compared to traditionally Ti implants [24,25]. Ti and zirconium (Zr) coatings are also used as an undercoating to improve the adhesion between the PEEK substrate and the HA coating [26,27].

In most studies, the authors report the deposition of coatings with layer thicknesses of more than 100 nm. However, ultra-thin coatings (thickness: <100 nm) on bone-integrating implants have the advantage of being able to maintain the topography of rough surfaces on sandblasted and/or acid-etched implants [28,29]. Therefore, even with 3D-printed porous structures, ultra-thin coatings can be used to preserve the original morphology and structure. In addition, the adhesion of thinner coatings is higher than that of thicker ones, as shown in Ref. [26].

In this work, the possibility of depositing ultrathin coatings (up to 100 nm) of Ti and Zr on polished disk-like PEEK substrates by direct current (DC) magnetron sputtering in an argon atmosphere was investigated. Both the ultra-thin Ti and Zr coatings were studied and compared with each other in terms of wettability, roughness, chemical composition and crystal structure. The modified PEEK surfaces were examined for changes in morphology and physicochemical properties.

## 2. Materials and Methods

### 2.1. Sample Preparation

In this work, PEEK (TECAPEEK, Ensinger GmbH, Nufringen, Germany) was used as disk-shaped samples with the following dimensions: 8.0 mm in diameter and 2.0 mm in height. In order to have a smooth surface and exclude the influence of different initial roughness, the samples were ground with P2000 sandpaper and polished with diamond suspension (grain size: 2–3 μm) on felt at a speed of 250.0 rpm for 30.0 min using a polishing machine (UNIPOL-802, Shenyang Kejing Auto-instrument Co., Ltd., Shenyang, China). Thereafter, the samples were cleaned using an ultrasonic bath (PSB-5735-05 Ultrasonic equipment, PSB-Gals, Moscow, Russian Federation), washing the samples with acetone (high purity grade, EKOS-1, Moscow, Russian Federation) first, then in isopropyl alcohol second (high purity grade, EKOS-1, Moscow, Russian Federation) and in distilled water for 10 min each third. After cleaning, the samples were placed in a vacuum oven (VTSH-K24-250, Aktan, Moscow, Russian Federation) at a temperature of 70.0 °C and a pressure of 0.5 Pa for 12 h.

### 2.2. Sample Modification

For the surface modification of the PEEK samples, coatings were prepared using a magnetron sputtering system (Katod-1M, Juterma, Rostov-on-Don, USSR) in direct current (DC) mode equipped with titanium and zirconium targets with a working area of 190 cm^2^, which was used to calculate power density below ((current·voltage)/working area). The vacuum chamber was first evacuated to a pressure of 7.0 × 10^−3^ Pa. Next, argon (Ar, 99.9999%, PTK Cryogen, Aramil, Russian Federation) was injected into the chamber until the chamber pressure was in the range of 0.7–0.9 Pa and this was maintained. A current of 2.0 A was applied for 10 min to clean both targets with a protective screen located above the targets. Subsequently, with the help of a rotating mechanism, a holder with samples was placed instead of the screen. To avoid overheating and thus damage to the PEEK samples, the magnetron sputtering process was carried out at a current value of 0.2 A for both the Ti (supporting information (SI) Appendix A) and Zr (SI Appendix A) targets. The power density was 126.0 mW/cm^2^ for Ti and 84.0 mW/cm^2^ for Zr. SI Appendix A shows the sputtering times for Ti and Zr, at which coatings with a thickness of 10 nm, 50 nm and 100 nm were obtained. The sample groups in this study are designated as follows: the unmodified PEEK samples as the control, the PEEK samples with Ti and Zr coatings are named according to the principle—coating element and then thickness (e.g., PEEK with a titanium coating of 10 nm thickness—Ti 10 nm).

### 2.3. Investigation Methods

To determine the coating thicknesses, Si wafers were placed in the sputtering chamber together with PEEK samples and then coated. The obtained coating thicknesses were determined using a spectral ellipsometer (Ellipse 1891 SAG, Scientific-Manufacturing Complex “Technological Centre”, Zelenograd, Moscow region, Russian Federation).

Surface topography and elemental composition were investigated via dispersive X-ray spectroscopy (EDX) and EDX mapping by scanning electron microscopy (SEM; Quanta 200 3D, FEI Company, Hillsboro, OR, USA) equipped with an EDX detector (Ametek EDAX, Mahwah, NJ, USA).

In order to characterize the three-dimensional morphology of the sample surfaces, an atomic force microscope (AFM; NT-MDT NTEGRA, Zelenograd, Moscow region, Russian Federation) operating in semicontact mode was used. A NSG01 AFM tip (NT-MDT NTEGRA, Zelenograd, Moscow region, Russian Federation) with an average force constant of 5.1 N/m was used to carry out these experiments. Arithmetic mean roughness (Ra) and average height difference (Rz) were calculated from AFM micrographs applying the software Gwyddion 2.60 (gwyddion.net, Brno, Czech Republic).

Wettability of the sample surfaces was carried out on a drop shape analyzer (Easy Drop DSA 20, Krüss, Hamburg, Germany) using the Drop Shape Analysis software, Version 1.92.1.1 (Krüss, Hamburg, Germany). To measure surface wettability, contact angles of water (Solopharm, Saint Petersburg, Russian Federation) and diiodomethane (99%, Acros Organics, Geel, Belgium) were determined. For this purpose, 3.0 µL drops of distilled water and diiodomethane were placed on the surface of each sample to be investigated. The measurements with water and diiodomethane were repeated five times for each sample to obtain the water contact angles (WCA) and diiodomethane contact angles (DCA). Surface energies were calculated using the Owens–Wendt–Rabel–Kaelble (OWRK) method.

Surface characterization was carried out using X-ray photoelectron spectroscopy (XPS; NEXSA, Thermo Fisher Scientific, Waltham, MA, USA) with a monochromated Al K alpha X-ray source operating at 1486.6 eV. Survey spectra were recorded at a pass energy of 200.0 eV, the high-resolution spectra at 50.0 eV, and with an energy resolution of 0.1 eV. The analyzed surface areas were 400.0 μm^2^. Investigations were carried out at room temperature in an ultra-high vacuum (UHV) with a pressure in the order of 1.0 × 10^−5^ Pa (in the case of use of an electron-ion compensation system, the argon partial pressure was 1.0 × 10^−3^ Pa).

Melting temperatures (*T_m_*) and melting enthalpies (Δ*H_fus_*) were determined via differential scanning calorimetry (DSC; SDT Q600, TA Instruments, New Castle, DE, USA). All DSC measurements were carried out in an argon atmosphere of 99.99% under the following conditions: a heating rate of 10 °C/min and a temperature range of 20 °C to 450 °C. The degrees of crystallinity *X_c_* were calculated using the following equation [30]:(1)$$X_{c}=\frac{\Delta H_{fus}}{\Delta H_{fus}^{0}}\;\cdot 100\%,$$
where $\Delta H_{fus}$ is the melting enthalpy of the sample and $\Delta H_{fus}^{0}$ is the melting enthalpy of an ideal crystal of PEEK. According to literature, the value of $\Delta H_{fus}^{0}$ is 130 J/g [30].

In order to evaluate the crystal structure of the coatings, samples with Ti and Zr coatings with a thickness of 600 nm were prepared. These coatings were evaluated by X-ray diffractometry (XRD; XRD 6000, Shimadzu, Kyoto, Japan) with a CuKα radiation source operating at a voltage of 40 kV and a current of 30 mA.

Figure 1 shows a detailed overview of the processes of sample polishing, the applied sample surface modification process and the investigation methods used to conduct this study.

## 3. Results and Discussion

### 3.1. Coating Thickness and Surface Morphology

As a result of the magnetron sputtering processes for titanium (Ti) and zirconium (Zr), the following deposition rates were obtained: 1.9 ± 0.2 nm/min for Ti (Supporting Information (SI) Appendix A) and 3.1 ± 0.7 nm /min for Zr (SI Appendix A). The coating thicknesses obtained (for the coating thicknesses to be achieved: 10 nm, 50 nm and 100 nm) are shown in SI Appendix A.

Figure 2 shows photographs, SEM and AFM micrographs of the control sample (unmodified) and samples surface-modified with titanium (Ti) and zirconium (Zr) coatings of different thicknesses. The surface-modified PEEK substrates retain their shape after magnetron sputtering, and all groups of samples have a smooth and shiny surface (Figure 2, left column). With increasing coating thickness, the color of the coating is changing from beige to violet color for Ti and pink for Zr. In the SEM and AFM micrographs (Figure 2, second and third columns), no obvious differences in the surface morphology of either the control samples or the samples with coatings can be observed; the surfaces are homogeneous with traces of polishing.

### 3.2. Elemental Composition of the Sample Surfaces

EDX analysis (Figure 3a) of the control sample shows that the surface consists of carbon (C) and oxygen (O) atoms, which are elements from the PEEK compound. In the case of the titanium (Ti) and zirconium (Zr) samples with a coating thickness of 10 nm, the elements Ti and Zr are now also observed in addition to C and O, and their content on the surfaces of the samples increases with increasing coating thickness (Figure 3a). Elemental mapping of the sample surfaces displays a uniform distribution of Ti or Zr on the respective polymer surfaces (SI Appendix A).

### 3.3. Surface Roughness

Based on AFM studies, the roughness parameters *R_a_* (arithmetic mean roughness) and *R_z_* (average height difference) of the control sample and samples with coatings were determined (Figure 3b). *R_a_* and *R_z_* of the control sample were 15 ± 1 nm and 67 ± 7 nm, respectively. As the coating thickness increases, *R_a_* and *R_z_* also increase. The maximum values of Ra and Rz were observed for Ti (27 ± 1 nm and 134 ± 18 nm, respectively) and for Zr (23 ± 4 nm and 139 ± 12 nm, respectively) with a coating thickness of about 100 nm. It should be noted that the increase in the roughness parameters with increasing deposition time can be explained by the formation of new islands of coating materials and an increase in their thickness [31].

### 3.4. Surface Wettability

Hydrophilic surfaces tend to enhance the early stages of cell adhesion, proliferation, differentiation, and bone mineralization compared to hydrophobic surfaces, so lower water contact angle (WCA) values are beneficial for osseointegration [32]. However, as shown in the literature [33], it is necessary to maintain a balance between hydrophilic and hydrophobic properties. Surfaces with high hydrophobic properties (WCA > 90°) show reduced cell affinity and thus reduced biocompatibility, while surfaces with high hydrophilic properties (WCA close to 0°) prevent intercellular interaction [34].

The unmodified control samples have a WCA of 76° ± 1° and a DCA of 19° ± 5° (Figure 3c). On the other hand, PEEK samples with a coating have a lower WCA (26–38% lower for Ti and 17–39% lower for Zr) compared to the WCA of the control samples. A different situation is observed with the diiodomethane contact angles (DCA). Compared to the control sample, the PEEK samples with coatings have higher DCA values (142–152% more for Ti and 168–194% more for Zr). In the case of Ti, the decrease in WCA is due to changes in the surface roughness [35]. The surface energy *γ* of the control samples is equal to 51.0 ± 1.0 mJ/m^2^ and has a predominantly dispersive character (Figure 3d). With an increase in the thickness of the Ti coatings, the surface energy increased from about 52.0 to 57.5 mJ/m^2^, mainly due to the increase in the polar component. With increasing Ti coating thickness, the ratio *γ^D^/γ^P^* decreases from ~ 2.1 for the Ti 10 nm PEEK samples to 1.7 for the Ti 100 nm samples. For Zr, the trend is slightly opposite. With increasing roughness, the surface energy *γ* of the Zr coatings decreases. In addition, the hydrophobicity of the Zr coatings increases with increasing coating thickness. There were similar observations for other Zr-based coatings and explained by an increase in covalent bonds with increasing coating thickness [36]. In the case of Zr 10 nm samples, *γ* sharply increases to 57.2 ± 0.8 mJ/m^2^ as compared to the control samples. The surface energy *γ* decreases to 45.4 mJ/m^2^ with increasing coating thickness. This decrease is mainly due to a decrease in the polar component *γ^P^* of *γ*.

### 3.5. X-ray Photoelectron Spectroscopy

SI Appendix A shows the survey XPS spectra of all investigated PEEK samples. The surface of the control sample shows O1s and C1s peaks, and the O/C ratio is 0.13, which is close to the literature value of 0.16 [37]. PEEK samples with Ti and Zr coatings have Ti2s, Ti2p, Zr3s, Zr3p, and Zr3d peaks, which are characteristic for these metals.

All high-resolution spectra (Ti2p, Zr3d and O1s) of the PEEK samples with different coating thicknesses were deconvoluted to determine the chemical composition of the coatings (Figure 4). For titanium, the XPS spectra of Ti2p (Figure 4, first column) are characterized by double peaks, Ti2p_1/2_ from 457.9 eV to 458.5 eV and Ti 2p_3/2_ from 463.6 eV to 464.2 eV, which correspond to the oxidation state of Ti^4+^ [38]. The high-resolution O1s spectra of the Ti coatings (Figure 4, second column) were decomposed into two XPS peaks; one peak is located from 529.4 eV to 529.9 eV, corresponding to the Ti-O bond, and the second peak at a higher binding energy of 531.0 eV is related to surface contamination [39]. These results confirm the presence of titanium oxide (TiO_2_). The XPS spectra of Zr3d consist of three peaks (Figure 4, third column). Two peaks correspond to the electron configuration Zr3d_3/2_ (183.9–184.1 eV) and to the electron configuration Zr3d_5/2_ (181.6–181.8 eV), and the third one corresponds to the shoulder in the region of 185.3–186.2 eV. The occurrence of the shoulder can be explained by defects in some crystal lattices of different zirconium compounds [39]. As with Ti, the O1s XPS spectra of Zr coatings are represented by two peaks (Figure 4, fourth column), one of which corresponds to the Zr-O bond (529.1–529.5 eV) [40], indicating the presence of ZrO_2_ on the surface, and the second peak corresponds to C-O/C=O bonds (531.0–531.3 eV). At the same time, no significant changes in the chemical structure depending on the thickness of the coatings on the PEEK samples were observed.

### 3.6. Crystal Structure of PEEK Substrate and Ti and Zr Coatings

The effect of the coating process on the PEEK substrate was evaluated using differential scanning calorimetry (DSC). The obtained DSC curves (Figure 5a) have a single endothermic peak at 340–343 °C, corresponding to the melting temperature (*T_m_*) of PEEK. Based on the degree of crystallinity of all PEEK samples, there is no significant difference in the crystal structure of PEEK with coatings compared to the control samples (Figure 5b). This proves that it is possible to deposit a coating on polymer substrates using the magnetron sputtering process without changing the crystal structure of the polymer backbone.

The X-ray diffraction patterns (XRD) show the crystal structure of the uncoated PEEK control sample and the surface-modified PEEK samples with Ti and Zr coatings of a thickness of 600 nm, which were specially prepared for coating structure evaluation (for more details, see chapter 2.3). (Figure 5c). In the XRD pattern of the unmodified control sample, three peaks are present in the region of 2*θ* = 18°–22°, which corresponds to the lattice planes (110), (113), and (200) of PEEK [41]. Moreover, the peak at 2*θ* = 29° is also in the lattice plane (213) that belongs to PEEK [41]. It should be noted that the surface-modified PEEK samples have the same peaks as the unmodified control samples, but no peaks corresponding to lattice planes of Ti or Zr can be identified. As a result, the XRD spectra obtained indicate amorphous Ti and Zr coatings formed on the surface of the PEEK samples, even at a coating thickness of 600 nm.

Physical and chemical surface modification methods are mainly strategies to improve osseointegration and cell adhesion of PEEK-based implants [10,42]. As a result of surface modification of PEEK samples with ultra-thin Ti and Zr-based coatings by DC magnetron sputtering, SEM micrographs and EDX analysis show uniformly distributed coatings over the PEEK surface with similar morphology to an uncoated PEEK sample. For both Ti and Zr based coatings, *R_a_* and *R_z_* increase with increasing coating thickness and remain in the nm range, opening the possibility of maintaining the topography and roughness of the initial PEEK surface when using different surface treatment methods. The advantage of a thin coating also lies in the flexibility of the coating [28], compared to thicker coatings [43]. However, Ti and Zr coatings exhibit different trends in the results. Despite the same trend in the morphology of both types of coatings (Figure 3b), with an increase in the thickness of the Zr coatings, the wettability changes towards hydrophobic, while the wettability of Ti coatings changes towards more hydrophilic. Based on the wettability results, the Ti and Zr coatings are the most hydrophilic at 100 and 10 nm thick, respectively (Figure 3c). By modifying PEEK with Ti and Zr coatings, it is possible to increase the surface energy of the polymer through the growth of the polar component and thus improve the hydrophilic properties of the PEEK substrate (Figure 3c,d). The results of XPS and XRD studies show that the coatings are represented by amorphous Ti oxides and Zr oxides, with Ti and Zr occurring primarily in their most stable oxidation state of +4 (Figure 4). Such materials are actively used as implants in the fields of orthopedics and dentistry [24,25,44,45]. Despite the fact that DC magnetron sputtering involves high heating of the PEEK substrate, the modified samples retained the original geometry as shown by the DSC results for the PEEK crystal structure (Figure 5a,b).

The coatings presented in this study show behavior in terms of wettability similar to that of the modification of PLLA-based samples with a titanium coating with a coating thickness of up to 10 nm [46]. As the authors note, titanium-modified PLLA surfaces demonstrate an increase in adhesion and proliferative activity in fibroblasts. In Ref. [47], the PEEK surface was modified using a sol–gel-derived TiO_2_ coating up to 30 nm thick. This coating also showed better adhesion of mesenchymal stem cells (MSCs) and their differentiation into osteoblasts. In in vivo tests, coated samples showed increased bone formation compared to uncoated PEEK [47]. PEEK implants coated with amorphous zirconium phosphate in nm thickness also show improved bioactive properties and an improved osseointegration compared to unmodified PEEK implants [28].

## 4. Conclusions

In this study, ultra-thin coatings (maximum layer thickness: 100 nm) of titanium and zirconium were deposited on polyether ether ketone-based samples using direct-current magnetron sputtering to improve the surface properties for potential biomedical applications. The fabrication of ultrathin titanium and zirconium coatings on the surface of polyether ether ketone-based substrates improved the wettability of the polymer surface. Water contact angles decreased by 26–38% for titanium coatings and 17–39% for zirconium coatings. In addition, with increasing thickness of the titanium coatings, the wettability of the surface changed toward being more hydrophilic, while that of zirconium coatings was more hydrophobic. At the same time, the surface morphology of all surface-modified samples was preserved. The geometry of the substrates and their surface structures also did not change because of the coatings. All coatings examined here consist of amorphous titanium and zirconium coatings, according to X-ray photoelectron spectroscopy results, while the surface layer consists of titanium oxide or zirconium oxide, according to X-ray diffraction results. Thus, the coatings presented in this study are promising with regard to their applications in the fields of orthopedics and dentistry.

## Figures and Tables

**Figure 1 materials-15-08029-f001:**
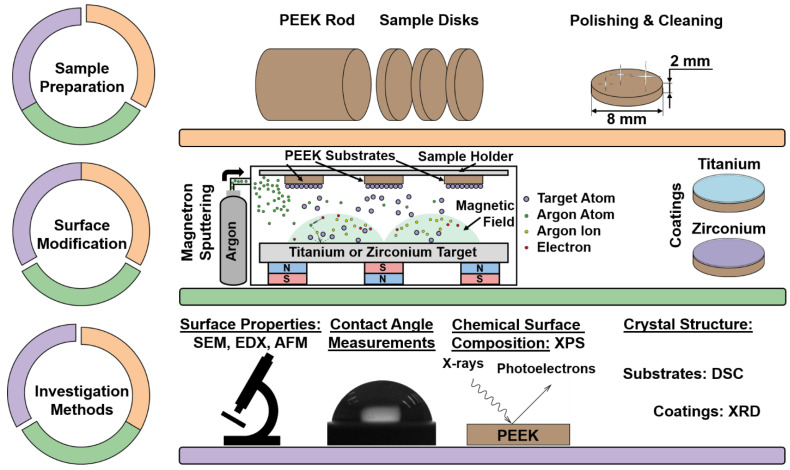
Schematic overview of the PEEK sample preparation process, the applied surface modification method, and the investigation methods used in this study.

**Figure 2 materials-15-08029-f002:**
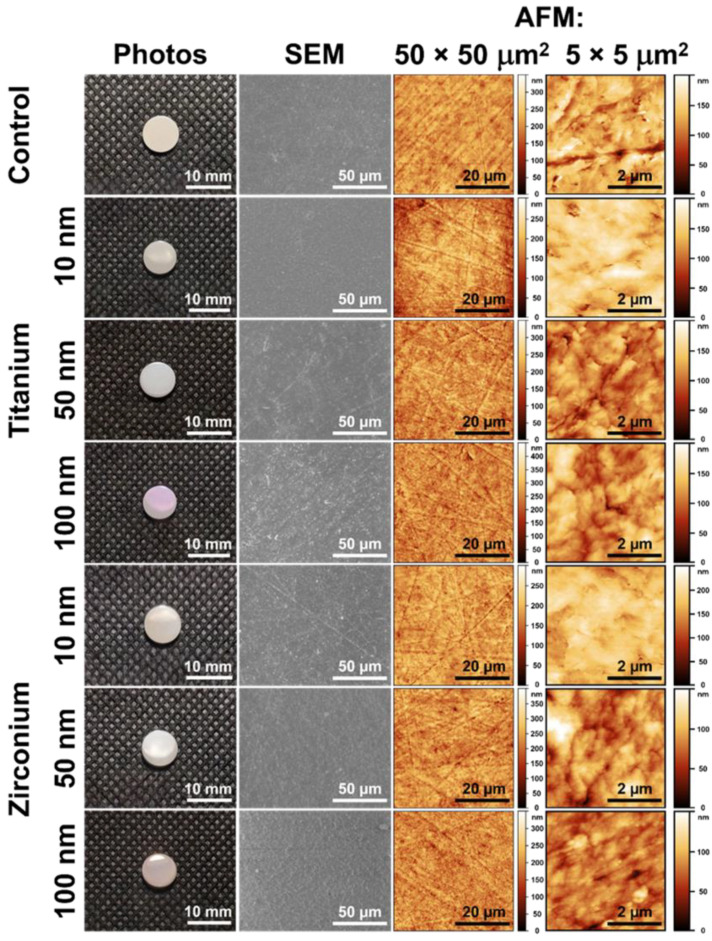
Photographs of the macroscopic appearance of all investigated sample surfaces (first column), SEM micrographs (second column), AFM micrographs of a surface area of 50 × 50 μm^2^ (third column) and AFM micrographs of a surface area of 5 × 5 μm^2^ (fourth column). The control sample is an unmodified PEEK sample surface and serves as a reference, the other samples are surface-modified with Ti and Zr coatings in thicknesses of 10, 50 and 100 nm.

**Figure 3 materials-15-08029-f003:**
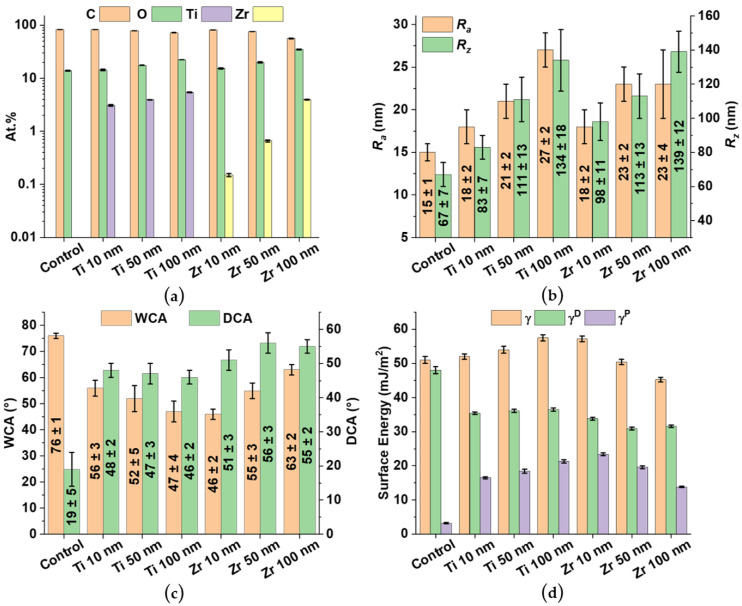
Chemical and surface properties of the investigated PEEK samples. Control is related to the unmodified PEEK samples, and Ti and Zr samples represent the ultra-thin film coated PEEK samples with coating thicknesses of 10 nm, 50 nm and 100 nm. (**a**) Elemental composition determined by EDX shown in at.% on a logarithmic scale for the y axis. (**b**) Surface roughness is given as *R_a_* (arithmetic mean roughness) and *R_z_* (average height difference) measured by AFM. (**c**) Surface wettability is presented as water contact angles (WCA) and diiodomethane contact angles (DCA). (**d**) The surface energy *γ* of the samples. Here, *γ* is also presented with the disperse component of the surface energy *γ^D^* and the polar component *γ^P^*.

**Figure 4 materials-15-08029-f004:**
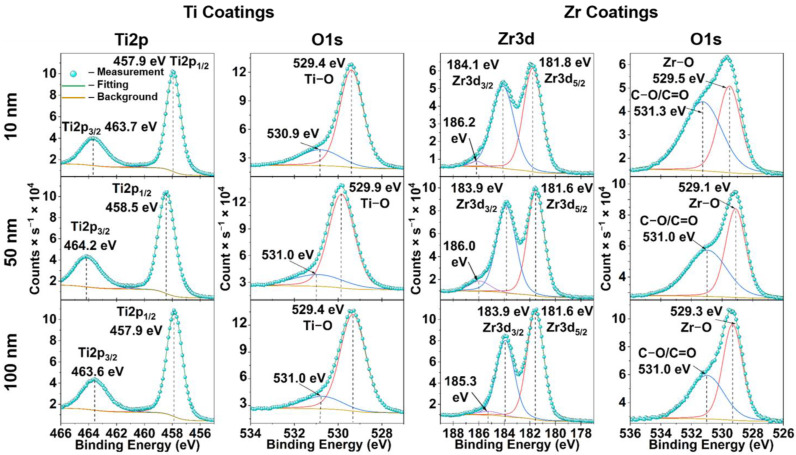
High-resolution XPS spectra of the obtained titanium (Ti) and zirconium (Zr) coatings with the thicknesses of 10 nm, 50 nm and 100 nm. Ti coatings are represented by Ti2p and O1s spectra; Zr3d and O1s spectra represent Zr coatings.

**Figure 5 materials-15-08029-f005:**
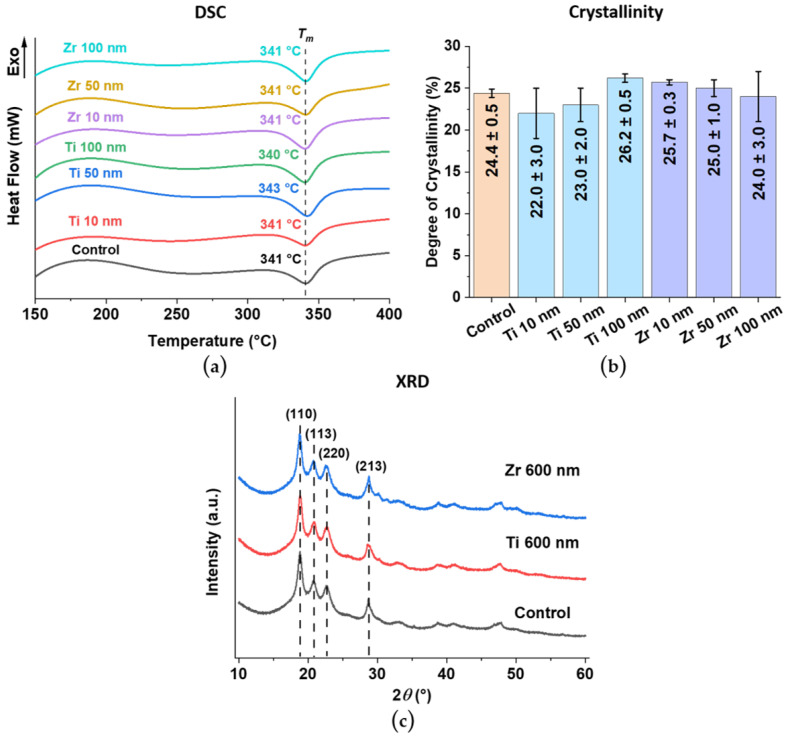
Crystallinity determination for all investigated PEEK samples. (**a**) Differential scanning calorimetry (DSC) thermograms with the melting temperature (*T_m_*) indicated. (**b**) Degree of crystallinity (*X_c_*) calculated from the DSC results using Equation (1) (see Chapter 2.3). (**c**) X-ray diffraction (XRD) spectra for unmodified control samples and surface-modified PEEK samples with a coating thickness of 600 nm.

## Data Availability

Data underlying the results presented in this paper are not publicly available at this time but may be obtained from the authors upon reasonable request.

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
