# Peer review of "Polyether Ether Ketone Coated with Ultra-Thin Films of Titanium Oxide and Zirconium Oxide Fabricated by DC Magnetron Sputtering for Biomedical Application"

_materials, 2022, doi:10.3390/ma15228029_

Round 1
Reviewer 1 Report
This is an interesting and topical manuscript on the surface modification of PEEK with sputtered Ti and Zr to improve its receptivity to human bone cells. Since the principal purpose of the research is to improve the attachment and proliferation of bone cells to PEEK, it is surprising to find that the study does not include any cell work. For completeness, the researchers should include a simple study to appraise the attachment of osteoblast or osteosarcoma cells to the original and modified PEEK samples. Alternatively, they could demonstrate that the Ti- and Zr-modified surfaces are more receptive to hydroxyapatite coatings than the pristine PEEK surface.
The title and information in the abstract and conclusion are misleading, as they suggest that the surface coatings comprise either elemental Ti or elemental Zr. XPS analysis demonstrates that the coatings are, in fact, principally composed of Ti or Zr oxides, and accordingly, the title, abstract and conclusion should reflect this.
The researchers should re-word or remove the following statement from the abstract, as it is purely speculative: ‘The coatings are suitable both as intermediate layers for hydroxyapatite coatings to improve adhesion and as separate coatings to improve the biocompatibility of polyether ether ketone based implants.’. They should also revise the similar statement in the conclusion.
Some factual information is not appropriately referenced (e.g., Lines 49 – 55). The researchers should either give the Powder Diffraction File number for the assignment of the XRD reflections of PEEK, or they should provide an appropriate reference.
The discussion is missing. Results are presented under the section entitled, ‘3. Results and Discussion’, but the researchers have failed to generate a contextual discussion of the significance of their findings.
Overall, the writing and presentation are of high quality and the manuscript is a pleasure to read. The minor grammatical, typing and formatting errors can be addressed at the copy editing stage. The references should be formatted in the mdpi style.
I have no confidential comments for the editors.
Author Response
Comment 1:
This is an interesting and topical manuscript on the surface modification of PEEK with sputtered Ti and Zr to improve its receptivity to human bone cells. Since the principal purpose of the research is to improve the attachment and proliferation of bone cells to PEEK, it is surprising to find that the study does not include any cell work. For completeness, the researchers should include a simple study to appraise the attachment of osteoblast or osteosarcoma cells to the original and modified PEEK samples. Alternatively, they could demonstrate that the Ti- and Zr-modified surfaces are more receptive to hydroxyapatite coatings than the pristine PEEK surface.
Answer: In general: We would like to thank the reviewer for his helpful comments and remarks.
We fully agree with this comment. A cell study would improve our article significantly. However, since we were invited to a special edition and unfortunately, we had a rather short period to conduct experiments and prepare the manuscript for submission. For this revision, we contacted our medical partners who are performing cell studies for us and spoke about a short performed research. For now, cell studies can be carried out only after 6 months due to the lack of required reagents, which must be imported. All chemicals are ordered, but the import is on long term now. However, the results of wettability shown in our manuscript reflecting a positive trend for the adhesion and proliferation of bone cells on the surface of the modified PEEK samples. Especially in case of titanium. In the last paragraph before the conclusion chapter, information has been added on articles with in vitro and in vivo studies using thin coatings to improve the surface properties of polymers.
Regarding the application of a hydroxyapatite layer on the surface of our modified samples, this is planned for a future paper, since our colleagues from the field of dentistry are interested in this research and an individual study plan is in process. Therefore, we ask for your understanding that in this study we do not want to add investigations on a hydroxyapatite layer on the surface of our modified samples.
Comment 2:
The title and information in the abstract and conclusion are misleading, as they suggest that the surface coatings comprise either elemental Ti or elemental Zr. XPS analysis demonstrates that the coatings are, in fact, principally composed of Ti or Zr oxides, and accordingly, the title, abstract and conclusion should reflect this.
Answer: We thank the reviewer for this comment. According to the comment, the manuscript title has been adjusted. The abstract and conclusion have been changed according to this comment (Lines 18 – 19, 361 – 363).
Comment 3:
The researchers should re-word or remove the following statement from the abstract, as it is purely speculative: ‘The coatings are suitable both as intermediate layers for hydroxyapatite coatings to improve adhesion and as separate coatings to improve the biocompatibility of polyether ether ketone based implants.’. They should also revise the similar statement in the conclusion.
Answer: We thank the reviewer for pointing out this issue. The statement has been removed from the abstract.
Comment 4:
Some factual information is not appropriately referenced (e.g., Lines 49 – 55). The researchers should either give the Powder Diffraction File number for the assignment of the XRD reflections of PEEK, or they should provide an appropriate reference.
Answer: We thank the reviewer for this useful comment. We added references on lines 52, 57, 307, 309 according to the comment.
Comment 5:
The discussion is missing. Results are presented under the section entitled, ‘3. Results and Discussion’, but the researchers have failed to generate a contextual discussion of the significance of their findings.
Answer: We thank the reviewer for this remark. The paragraph after Figure 5 and before Section 4 has been restructured and discussions improved.
For explanation: The main goal of this work was to enhance the hydrophilic properties of the polymer by applying ultra-thin coatings to improve PEEK in terms of biocompatibility and osseointegration. In the framework of the investigations performed, we focused on the wettability and surface energy, which we were able to increase through the polar component. Comparing our results with those of the papers cited in the paragraph before section 4, ultra-thin coatings exhibit similar behavior. In the future, we plan to conduct in vivo and in vitro studies with the here investigated ultra-thin titanium and zirconium based coatings.
Comment 6:
Overall, the writing and presentation are of high quality and the manuscript is a pleasure to read. The minor grammatical, typing and formatting errors can be addressed at the copy editing stage. The references should be formatted in the mdpi style.
I have no confidential comments for the editors.
Answer: We would like to thank the reviewer for his opinion and judgement.

Reviewer 2 Report
See the attached pdf file of my report.

Author Response
The topic of the article is relevant and concerns an important area of research on biocompatible materials for medical applications. The authors investigate the properties of nanosized Ti and Zr films DC magnetron sputtered onto Polyether ether ketone (PEEK) and the influence of these films on the surface properties of the substrate. The article is well written, easy to read, the results from the extensive research applying AFM, SEM, EDX, XRD, XPS, DSC, etc., are well presented and discussed. The results and discussions are supported by adequate references. The supplementary materials are appropriately selected to shed light on some results. My opinion is that the present manuscript can be accepted for publication after a minor revision.
Answer: We would like to thank the reviewer for his judgement and helpful remarks.
I have some remarks which should be considered by the authors:
Abstract:
- Page 1, line 17; The verb is missing in the sentence “Investigation results showing an even distribution........”. I suggest: “Investigation results show a uniform .."
Answer: We thank the reviewer for this pointing out this issue. The sentence has been adjusted according to this comment.
- Page 1, lines 21-22; The sentence “These coatings improve wettability…..” should be revised. I suggest : “These coatings improve wettability increasing surface energies, in particular increasing the polar component, which in turn is important for cell adhesion.
done
Answer: Thanks to the reviewer. The sentence has been adjusted.
Section 2.1:
- Page 3, line 98; Why do you need such a long (12 hours) drying process of PEEK substrates, even more so in a residual gas atmosphere (0.5 Pa)?
Answer: After polishing, the PEEK samples were cleaned in organic solvents and in water. Such a long time and a pressure of 0.5 Pa was chosen to ensure that all residual solvents and water were removed from the samples before they were placed in the vacuum chamber of the magnetron sputtering system.
Section 2.2:
- Page3, lines 108-109, I assume that during the cleaning of the targets, the authors use a screen that protects the samples. This should be clarified in the article.
Answer: We thank the reviewer for pointing out this aspect. Yes that is correct, such a screen was used. This point has been added and clarified in chapter 2.2 of the main manuscript.
Section 2.3:
- In the article, pressure units such as Pa and mbar are used. However, it is desirable to use the same units for physical quantities throughout the article (in the SI system this unit is Pa).
Answer: We thank the reviewer for pointing out this issue. The pressure is now used exclusively in the unit Pa.
- Page 4, line 164; Fig. 1 should be mentioned also in the text (after line 164) adding a sentence describing the contents of Fig. 1.
Answer: We thank the reviewer for this point. We totally agree with the reviewer and apologize that this was not already present in the first submitted variant of the manuscript. A corresponding sentence has been added before Figure 1.
Section 3.1:
- Page 4, lines 172-175; The deposition rate of Zr (given as 3.1 ± 0.7 nm/min) is greater than that of Ti (given as 1.9 ± 0.2 nm /min). However, calculation of the rate from each experimental point in Fig.S1c,d gives the following results: for Ti: ~ 10/4 = 2.5, ~ 50/20 = 2,5 and ~ 100/50 = 2 nm/min; For Zr: ~ 10/2.5 = 4, ~ 50/10 = 5 and ~ 100/30= 3.3 nm/min. Furthermore, the slope of the curves in both Figures S1c and S1d is the same, given as 1.9 ± 0.2 nm /min. What are the actual deposition rates of these metals?
Answer: We thank the reviewer very much for finding this issue! Something went wrong, we sincerely apologize. The figure now shows the correct values for the slope of the curve in SI Figure S1d. The deposition rate was determined by the slope of the straight line obtained by a three-point linear fit of the coating thickness and deposition time dependence.
- Page 5, Fig. 2 and line 191; The unit of scanned area in AFM images is μm2, adjust this in Fig. 2 and in the caption to it (50x50 μm2; 5x5 μm2).
Answer: Thanks to the reviewer for pointing out this issue. Figure 2 and the corresponding figure caption has been adjusted according this comment.
Section 3.4:
- Page 7, Fig. 3, lines 244-246; In the caption of Fig. 3a and 3b the word “and” is superfluous, it should be deleted: (a).......” determined by EDX and shown in at.% on” and (b) ... “Rz (average height difference) and measured by AFM.”
Answer: We thank the reviewer for finding this mistake. The figure caption has been corrected.
Section 4:
- Based on the XPS results, the authors state that “The coatings are represented by amorphous titanium dioxide and zirconium dioxide uniformly distributed over the surface of the polyether ether ketone substrates. “ In general, the XPS signal comes from a few nm thick surface layers, so it is reasonable that the thinnest layers are Ti and Zr oxides. But what about the thicker layers (50, 100 nm) do they also oxidize in depth?. Have you done a depth profile to verify this?
Answer: We thank the reviewer for these questions. According to this comment, the sentence in the conclusion was reformulated. Indeed, in the framework of this study, we did not study coatings in depth. Please note that we were invited to a special edition of this journal and unfortunately, we had a rather short period to conduct experiments and prepare the manuscript for submission. As an general answer: According to previous studies we conducted, we can assume that a certain amount of metal oxides were formed directly on the surface of the sample due to some oxygen molecules coming from the backbone of the used polymer substrate. The metal oxide compounds that formed on the coating surface and determined by XPS most likely formed with the residual oxygen in the Magnetron sputtering chamber. Therefore, we also assume that the middle part of the coatings is mainly composed of amorphous metal coatings, as we can see from the XRD results.

Reviewer 3 Report
This study presents a study of ultra-thin films of titanium and zirconium fabricated by DC magnetron sputtering for biomedical application. Since, some of the claims lack proof and this gives rise to question about their results. Some recommendation for improving the article as follows:
1. The title of the manuscript is ultra-thin films of titanium and zirconium for biomedical application, however I don’t find any studies on the applications. I suggest the authors discuss the details of the properties related to the applications add more references.
2. I don’t think 100 nm thick Ti and Zr thin films is thick enough for biomedical application.
3. Would you please explain the reason of the higher Ra with Ti and Zr coatings? How does the rougher surface help to improve the properties of the Polyether ether ketone?
4. Based on your studies Ti films shows better properties than the Cr films, is there any merits in Zr coatings? Would please point out the optimum thickness of Ti and Zr coatings?
Author Response
This study presents a study of ultra-thin films of titanium and zirconium fabricated by DC magnetron sputtering for biomedical application. Since, some of the claims lack proof and this gives rise to question about their results. Some recommendation for improving the article as follows:
- The title of the manuscript is ultra-thin films of titanium and zirconium for biomedical application, however I don’t find any studies on the applications. I suggest the authors discuss the details of the properties related to the applications add more references.
Answer: We thank the reviewer very much for his comments and remarks. Before section 4, information were added on articles with in vitro and in vivo studies in which thin coatings were used to improve the surface properties of polymer implants
- I don’t think 100 nm thick Ti and Zr thin films is thick enough for biomedical application.
Answer: We thank the reviewer for this point. The topic of thin coatings for biomedical applications is the subject of a large number of studies, articles, reviews, and books. An example of one of these books, which shows various methods for applying coatings of various thicknesses:
Griesser, Hans J., ed. "Thin film coatings for biomaterials and biomedical applications." (2016).
Also, in one of the added sources cited in the article, the PEEK surface was modified with sol-gel derived TiO2 up to 30 nm thick. The paper presents in vitro and in vivo proving the effectiveness of such coatings:
Shimizu, Takayoshi, et al. "Bioactivity of sol–gel-derived TiO2 coating on polyetheretherketone: In vitro and in vivo studies." Acta biomaterialia 35 (2016): 305-317.
In addition, ultra-thin coatings were also requested by some medical cooperation partners (especially dentists) who want to carry out studies with our coatings presented here as part of their research projects.
- Would you please explain the reason of the higher Ra with Ti and Zr coatings? How does the rougher surface help to improve the properties of the Polyether ether ketone?
Answer: We thank the reviewer for these questions. As we have already noted in the main manuscript, the change in roughness is caused by the formation of islands on the surface and a change in their height. The mechanism of formation and growth of thin films is described in more detail in the work of Petrov, I., et al. "Microstructural evolution during film growth." Journal of Vacuum Science & Technology A: Vacuum, Surfaces, and Films 21.5 (2003): S117-S128. The improvements for the PEEK surfaces make sense from a cell adhesion point of view, depending on the cell type, a certain range of roughness promotes cell adhesion. The surfaces should not be too smooth, but not too rough either. The island formation of the coatings described above is also an advantage, since cells (especially osteoblasts and fibroblasts) like to settle in areas that initially offer a kind of support and protection.
- Based on your studies Ti films shows better properties than the Zr films, is there any merits in Zr coatings? Would please point out the optimum thickness of Ti and Zr coatings?
Answer: We also thank the reviewer for these questions. Various in vivo and in vitro soft tissue response studies around zirconia have shown comparable or even better healing response, less inflammatory infiltrate, and reduced plaque adhesion on zirconia discs compared to conventionally pure titanium.
Osman, Reham B., and Michael V. Swain. "A critical review of dental implant materials with an emphasis on titanium versus zirconia." Materials 8.3 (2015): 932-958.
Sivaraman, Karthik, et al. "Is zirconia a viable alternative to titanium for oral implant? A critical review." Journal of Prosthodontic Research 62.2 (2018): 121-133.
However, most clinical studies of zirconia based implants are short term and the limiting factor is how they perform in the long term. We plan to continue working with both types of coatings and in the near future to conduct in vitro and in vivo studies and compare the effects of Ti and Zr on the PEEK surface. We have also added a sentence on the optimal thickness of Ti and Zr coatings according to the results of this study on page 11 of the main manuscript.

Round 2
Reviewer 1 Report
The authors have adequately addressed all major concerns and I recommend that the paper be accepted, subject to minor Grammatical corrections by the copy editor.
I have no confidential comments for the editors.
Reviewer 3 Report
Accept as it is.